# The Distributional Hypothesis Does Not Fully Explain the Benefits of Masked Language Model Pretraining

**Ting-Rui Chiang** and **Dani Yogatama**
University of Southern California
{tingruic,yogatama}@usc.edu

## Abstract

We analyze the masked language modeling pretraining objective function from the perspective of the distributional hypothesis. We investigate whether better sample efficiency and the better generalization capability of models pretrained with masked language modeling can be attributed to the semantic similarity encoded in the pretraining data's distributional property. Via a synthetic dataset, our analysis suggests that distributional property indeed leads to the better sample efficiency of pretrained masked language models, but does not fully explain the generalization capability. We also conduct analyses over two real-world datasets and demonstrate that the distributional property does not explain the generalization ability of pretrained natural language models either. Our results illustrate our limited understanding of model pretraining and provide future research directions. [1]

## 1 Introduction

Despite the rise of the prompting paradigm with the scaling breakthrough of very large language models (Brown et al., 2020), understanding the mechanism of model fine-tuning remains to be an important endeavor. Fine-tuning relatively small models such as BERT (Peters et al., 2018; Devlin et al., 2019; Liu et al., 2019b) has significant practical implications since these models are computationally more efficient than large language models such as GPT-3 (Brown et al., 2020). Distilling a large language model to a small model by fine-tuning is also a promising direction (Lang et al., 2022; Hinton et al., 2015). The development of large language models also involves fine-tuning, e.g. doing reinforcement learning from human feedback (Bai et al., 2022). In this work, we seek to understand how pretraining benefits downstream fine-tuning.

Previous analyses of pretrained model focus on probing the pretrained representations (Tenney et al., 2019b,a; Liu et al., 2019a; Hewitt and Manning, 2019; Wu et al., 2020; Rogers et al., 2020; Zhang et al., 2021; Immer et al., 2022). However, these models are rarely used as a static feature extractor. Practitioners often fine-tune them with downstream tasks.

We analyze whether mask language model pretraining allows models to leverage semantic knowledge derived from the distribution of words in the pretraining data for downstream tasks. This research question follows the distributional hypothesis (Harris, 1954; Firth, 1957). It postulates that semantically related words have a similar context distribution, which we elaborate more rigorously as a distributional property of pretraining data in §2. Because it draws connections between word semantics and data distribution, it has been widely accepted as an explanation for the efficacy of non-contextualized (type-based) word embeddings (Levy and Goldberg, 2014) such as Word2Vec (Mikolov et al., 2013) and GloVe (Pennington et al., 2014). Recently, Sinha et al. (2021) shows that word order does not greatly affect the efficacy of masked language model pretraining (Devlin et al., 2019; Liu et al., 2019b), thus conjecturing that the distributional hypothesis could be an explanation. In this work, we continue studying this conjecture by investigating whether pretrained models can utilize the semantic knowledge derived from the data distribution as suggested by the distribution hypothesis.

Among different types of semantic knowledge the distribution of the pretraining data may encode, we focus on semantic equivalence, i.e., synonym, which Harris (1954) suggests to be encoded in the data distribution. In §3, we theoretically show that prior knowledge about semantic equivalence alone is enough to lead to two desirable properties that pretrained language models have: better sample

---

[1] Scripts for the experiments in this paper are available at https://github.com/usc-tamagotchi/DH-MLM.

efficiency and generalization capability (Tu et al., 2020; Hendrycks et al., 2020; Tänzer et al., 2022).

In §4, as our first step to understanding how masked language modeling training objective exploits the distributional property of the pretraining data, we set up a toy experiment. We construct a synthetic pretraining corpus and a downstream task. In §5, we show that the distributional property explains the sample efficiency of pretrained masked language models, but it does not always lead to better generalization on its own; it still depends on the type of distribution shifts. Our results also suggest that better parameter initialization statistics (Wu et al., 2022) do not explain the benefit of masked language model pretraining either.

In §6, we conduct a similar experiment in the real-world setting. We analyze a pretrained BERT model and two real-world tasks SST-2 (Socher et al., 2013; Wang et al., 2018) and MNLI (Williams et al., 2018). We find that the semantic (in)equivalence the model learns from the pretraining task is independent of how a fine-tuned model treats words as synonyms. This indicates that pretrained models do not generalize better by modeling the distributional property of the pretraining data either.

To summarize, our main findings include: i) We show that semantic equivalence encoded in the distributional property of the pretraining data makes pretrained models more sample-efficient. ii) There is a type of generalization capability independent of the distributional property of the pretraining data. iii) The distributional property of the pretraining data does not explain pretrained models' generalization capability in the real world. Therefore, we conclude that the distributional hypothesis by Harris (1954) alone is not enough for a complete explanation. Future work may study the interplay between other complex semantic relationships, data distribution, and model pretraining to better understand pretrained models' generalization behavior.

## 2   Properties of Pretraining Data and Downstream Tasks

Since we focus on semantic equivalence in language, we use the concept of "synsets" from Word-Net (Miller, 1998), i.e., the sets of synonyms. Here we extend the concept of synsets to allow each synset to contain multi-token elements, such as phrases or sentences. Elements in a synset are *features* and each of them is a sequence of tokens (or

a single token). Having these semantic equivalence sets has the following implications:

**In the pretraining data:**   The distributional hypothesis suggests that the pretraining data has a distributional property: for any two features $a, b$ in the same synset and any $n \geq 1$, the distribution of their neighboring words satisfies

$$p(x_1, x_2, \cdots, x_n|a) \approx p(x_1, x_2, \cdots, x_n|b). \quad (1)$$

For example, the phrase "is delicious" has the same meaning as the phrase "taste good". Therefore, if the training data has this distributional property, then we expect its distribution to satisfy

$$p(x|\text{"is delicious"}) \approx p(x|\text{"tastes good"}).$$

**In a downstream task:**   Given an input $\boldsymbol{x}$ for a downstream task, substituting a feature $a$ with another feature $b$ in the same synset does not change the meaning of $\boldsymbol{x}$ because $a$ and $b$ are synonyms. Therefore, the substitution does not change the label of $\boldsymbol{x}$ either:

$$f^*(\boldsymbol{x}) = f^*(\text{Replace}(\boldsymbol{x}, a, b)), \quad (2)$$

where $f^*$ is the task's labeling function that maps an input $\boldsymbol{x}$ to its label $y$.

## 3   The Benefit of Modeling Semantic Equivalence

In this section, we show why having prior knowledge about the semantic equivalence sets (synsets) helps downstream performance.

### 3.1   Sample Efficiency

Understanding which symbols are semantically related makes a classification problem easier. From the perspective of the model, the inputs are sequences of symbols. Having prior knowledge about the relationship between these symbols decreases the number of training examples the model requires to learn the target labeling function.

For example, consider a task with four training examples: ⟨"It is delicious", True⟩, ⟨"It is awful", False⟩, ⟨"It is tasty", True⟩, ⟨"It is bad", False⟩. If the model does not know the semantic relationship between the predictive features "awful", "bad", "tasty", and "delicious", then it is impossible for the model to learn the underlying target labeling function from these four samples, because each of these four words appears only once in the dataset.

Pre-define two Markov chains as $P_\Sigma^{(1)}$ $P_\Sigma^{(2)}$

1. Choose $P_\Sigma^{(1)}$ or $P_\Sigma^{(2)}$

2. Sample synsets

$P_\Sigma^{(1)}$ —— sample
$\sigma_1\ \sigma_2\ \sigma_3\ \sigma_1\ \sigma_2$

$P_\Sigma^{(2)}$ —— sample
$\sigma_3\ \sigma_2\ \sigma_1\ \sigma_3\ \sigma_2$

3. Sample features

$a_1\ a_2\ a_3\ a_1\ a_2$
$b_3\ b_2\ b_1\ b_3\ b_2$
not satisfying DH

$a_1\ b_2\ a_3\ b_1\ a_2$
$a_3\ a_2\ b_1\ a_3\ b_2$
satisfying DH

4. Map features to tokens

| | | |
|---|---|---|
| $a_1 \to \alpha_1$ | $a_1 \to \alpha_1\alpha_3$ | $a_1 \to \alpha_1\alpha_1$ |
| $b_1 \to \beta_1$ | $b_1 \to \beta_2$ | $b_1 \to \alpha_2$ |
| ... | ... | ... |
| single-token | mulit-token w/o vocab. sharing | mulit-token w/ vocab. sharing |

Figure 1: The generation process of the pretraining data (§4.1). The words in the orange color denote independent experimental variables.

However, the knowledge that some symbols satisfy Eq. 2 makes the task easier, e.g. $f(\text{"It is delicious"}) = f(\text{Replace("It is delicious", "delicious", "tasty")})$. In other words, it reduces the feature space. Based on statistical learning theory (Valiant, 1984; Blum and Mitchell, 1998), this smaller feature space reduces the sample complexity of the model.

## 3.2 Generalization Capability

If a pretrained model is able to preserve its knowledge about semantic equivalence among words after being fine-tuned, then it will help the model to generalize better. For example, consider a simplified case with two training examples ⟨"It is delicious", True⟩ and ⟨"It is awful", False⟩, as well as a test example "It is tasty". Generalizing to this test example is possible only when the model understands the semantic relationships between "delicious", "awful" and "tasty". That is, if a model has an inductive bias in Eq. 2, as long as $\boldsymbol{x}$ containing $a$ is in the training set, the model will be able to generalize to a testing sample $\text{Replace}(\boldsymbol{x}, a, b)$, where $b$ is an unseen feature semantically equivalent to $a$.

## 4 Synthetic Data Construction

## 4.1 Pretraining Datasets

For pretraining, we construct a pseudo-language with the properties described in §2. This language has $n$ synsets $\Sigma = \{\sigma_1, \sigma_2, \cdots, \sigma_n\}$ and each

synset contains two features:

$$s_i = \{a_i, b_i\}.$$

This provides us with two semantically isomorphic sets of features $\Phi_a = \{a_i\}_{i=1}^n$ and $\Phi_b = \{b_i\}_{i=1}^n$. Each feature analogizes to a word or a phrase in natural languages, depending on whether each feature corresponds to one or multiple tokens. We discuss the multi-token setting in A.1.

To understand how masked language model pretraining would help models utilize the semantic isomorphism between $\Phi_a$ and $\Phi_b$, we start with a setup where the semantic isomorphism is encoded with a simple distribution that language models can learn easily. Specifically, we first randomly generate two Markov chains $P_\Sigma^{(1)}, P_\Sigma^{(2)}$ and use them to generate sentences (Figure 1):

1. We randomly choose $k$ from $\{1, 2\}$, which decides whether to use the first Markov chain $P_\Sigma^{(1)}$ or the second one $P_\Sigma^{(2)}$.

2. We then draw a sequence of synsets $s_1, s_2, \cdots, s_l$ based on the distribution defined by the chosen Markov chain $P_\Sigma^{(k)}(s_1, s_2, \cdots, s_l)$.

3. Then for $i = 1, 2, \cdots, l$, we draw a feature $x_i \in s_i$. At this step, we can control the distributional property as an independent experimental variable. We can generate a dataset with the distributional property in Eq. 1 by drawing features from $s_i = \{a_i, b_i\}$ uniformly at random. Or we can generate a dataset without the distributional property by always drawing $a_i$ or $b_i$ when $k = 1$ or $k = 2$ respectively.

4. Finally, we map each feature to a single token. (In A.1, we describe multi-token setups where we map each feature to multiple tokens.)

## 4.2 Downstream Task

We construct a synthetic downstream task aligned with the semantic equivalence specified in Eq. 2. We define a pattern-matching task where the labeling function that maps a sequence of features to a label based on the underlying synsets sequence that generates the feature sequence. If the sequence of synsets matches one of a predefined set of patterns, then we label the sequence as positive and otherwise negative.

We define these patterns with regular expressions in this form:

$$\Sigma^* \ S_1 \ \Sigma^* \ S_2 \ \Sigma^* \ S_3 \ \Sigma^*,$$

where $\Sigma = \{\sigma_1, \sigma_2, \cdots, \sigma_n\}$, $S_1, S_2, S_3 \subset \Sigma$ are 3 randomly selected sets of synsets and $*$ is a Kleene star. In other words, a sequence of synsets matches the pattern if synsets in $S_1, S_2, S_3$ appear in the sequence in order.

We generate *four* downstream datasets as follows. First, we can choose to generate the sequence of synsets from $P_\Sigma^{(1)}$ or $P_\Sigma^{(2)}$. We will use D1 and D2 to denote the samples generated from these two distributions respectively. Second, for each synset $s_i$, we can choose to always use features in $\Phi_a$ or $\Phi_b$. We will use A and B to denote the samples constituted with these two feature sets respectively. Therefore, there are 4 combinations of choices: A-D1, B-D1, A-D2, and B-D2. We will use these 4 combinations to test the generalization capability and the sample efficiency of the pretrained models.

## 5 Synthetic Experiment Results

### 5.1 Experiment Design

We first generate two pretraining datasets of the same size, one of which has the distributional property in Eq. 1 while the other one does not (according to §4.1). We then use these two datasets to pretrain two Transformer models (Vaswani et al., 2017) with the masked language modeling objective (the w/ DH and w/o DH model). Finally, we fine-tune these two models with a downstream task dataset generated in §4.2 (more details in B). We report the performance of the models after fine-tuning them with different numbers of downstream examples. By comparing the performance of the downstream task, we can infer to what extent the distributional property contributes to the efficacy of MLM pretraining. Despite the complexity of natural language, we posit that a portion of semantic equivalence in natural language is encoded in a distribution as simple as the one of this pseudo language. Therefore, positive results observed in this toy experiment would also apply to real-world setups. In the following, we describe how we use the four downstream datasets (i.e., A-D1, B-D1, A-D2, and B-D2) for evaluation:

**Sample efficiency.** We fine-tune the models with a dataset in which 50% of examples are in A-D1 and the other 50% are in B-D2. In this setting, if

the model has the knowledge about the semantic isomorphism between $\Phi_a$ and $\Phi_b$, then the model will be able to use the examples from both domains to learn a single labeling function. However, if the model has no such prior knowledge, then it will need to learn two labeling functions for A-D1 and B-D2 respectively with only half of the total amount of data. Thus, if the model can learn to utilize the semantic equivalence encoded in the distribution property (Eq. 1) of the pretraining data, then models pretrained with dataset having the property in Eq. 1 will perform better.

**Generalization capability.** We assess generalization capability by evaluating the model on a test set whose distribution is different from the model's training set. Specifically, we fine-tune the model with a training set in A-D1 and test it with test sets in all of the 4 distributions. We also explore whether having a small amount of data from a target domain can improve the generalization. Thus we also experiment with the setting where 10% of the training samples are from B-D2.

There are distribution shifts in two directions in this setting. One direction is the distribution shift from feature set $\Phi_a$ to $\Phi_b$ (e.g. from A-D1 to B-D1). We refer to this direction as a *vocabulary shift*, because the vocabulary is changed. The other direction is the shift from $P_\Sigma^{(1)}$ to $P_\Sigma^{(2)}$ (e.g. from A-D1 to A-D2). We refer to it as a *semantic shift*, because the correlation of the synsets is changed.

These two distribution shifts exist in real-world NLP scenarios. The vocabulary shift is analogous to cross-lingual generalization. It is similar to fine-tuning a multi-lingual pretrained model with a language and doing zero-shot transfer to a test set in another language (Artetxe et al., 2020). As for the semantic shift, it is related to the spurious correlations in training data (Poliak et al., 2018; Gururangan et al., 2018; McCoy et al., 2019; Chiang et al., 2020; Gardner et al., 2021; Eisenstein, 2022).

### 5.2 Other Baselines

In addition to the model pretrained with a dataset that does not have the distributional property (Eq. 1), we also have other baselines:

- From-scratch: We fine-tune the model from scratch with randomly initialized parameters.

- CBOW: We pretrain CBOW embeddings with the data that has the distributional property

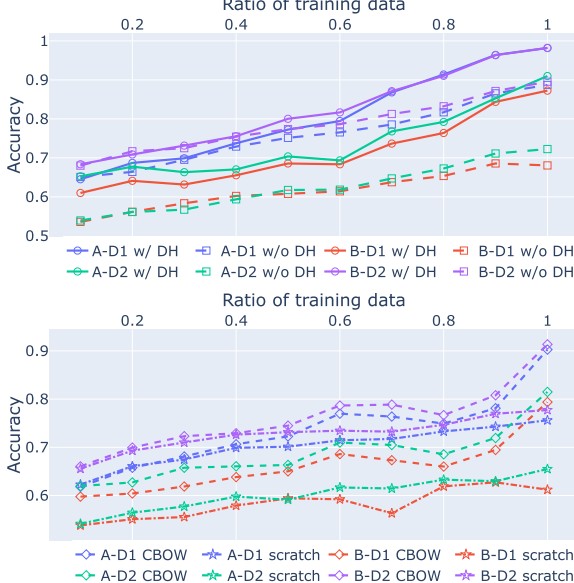

Figure 2: Performance on the synthetic downstream task when the models are fine-tuned with 50% A-D1 and 50% B-D2 (§4.2).

and use it to initialize the embedding layer of a randomly initialized Transformer.

- Shuffle-weight: We randomly shuffle the weights in each module of the MLM pretrained model before fine-tuning it, which keeps the statistics of the parameters. This is motivated by a previous work showing the efficacy of parameter initialization statistics (Wu et al., 2022). We inspect whether the statistics explain the efficacy of the model.

## 5.3 Experiment Results [2]

### 5.3.1 Sample Efficiency

We find that the distributional property (Eq. 1) can largely explain MLMs' better sample efficiency. In Figure 2, we can see that the w/ DH model's performance on A-D1 and B-D2 grows faster than the performance of the w/o DH model. This implies that pretraining with data satisfying Eq. 1 indeed improves the sample efficiency. The w/ DH model also has better performance on B-D1 and A-D2. It is aligned with our intuition that Eq. 1 can help the model utilize the isomorphism between $\Phi_a$ and $\Phi_b$ and learn a single general function instead of two functions for A-D1 and B-D2.

---

[2]Models initialized with shuffled weights have similar performance as models trained from scratch so we omit them in the figures for clarity.

The performance of the CBOW initialization is also better than training from scratch. Thus, the distributional property in Eq. 1 is indeed a valid explanation for non-contextualized embeddings. However, its improvement is not as stable as the w/ DH pretrained MLM. It suggests that in addition to the property of distribution, some other factors also attribute a model's better sample efficiency.

### 5.3.2 Generalization Capability

**Generalization in the presence of vocabulary shifts.** The results in Figure 3a and 3b indicate that the distributional property in Eq. 1 can help the generalization when there are vocabulary shifts to some extent. We observe that the w/o DH model does not generalize to B-D1 and B-D2 at all, while the w/ DH model can generalize from domain A-D1 to the test set in B-D1 and B-D2. However, the generalization diminishes as the amount of fine-tuning data increases. The model may suffer from catastrophic forgetting after we fine-tune it more.

Having some fine-tuning data in B-D2 mitigates this problem. As shown in Figure 3b, when the 10% of the fine-tuning data is in B-D2, the w/ DH model outperforms the w/o DH model much for the test sets in B-D1 and B-D2, and the generalization does not diminish.

The CBOW model also generalizes to B-D1 and B-D2. Its generalization to B-D1 does not suffer from catastrophic forgetting. Thus, the distribution property in Eq. 1 indeed largely explains the efficacy of non-contextualized word embedding.

**Generalization to the semantic shifts.** We find that the distributional property in Eq. 1 can not explain it. From both Figure 3a and Figure 3b, we can see that both the w/o DH model and the w/ DH model generalize to A-D2 better than the model trained from scratch. Moreover, the w/ DH model does not perform better. Meanwhile, the CBOW model performs only slightly better than training from scratch. Thus, generalization to the semantic shifts may be related to the nature of MLM pretraining instead of data properties (Chiang, 2021).

## 5.4 Summary

Our results suggest that pretraining with data whose distribution encodes the semantic equivalence between words (Eq. 1) improves pretrained MLM's sample efficiency, but the generalization capability in the presence of vocabulary shifts diminishes as the number of training examples grows.

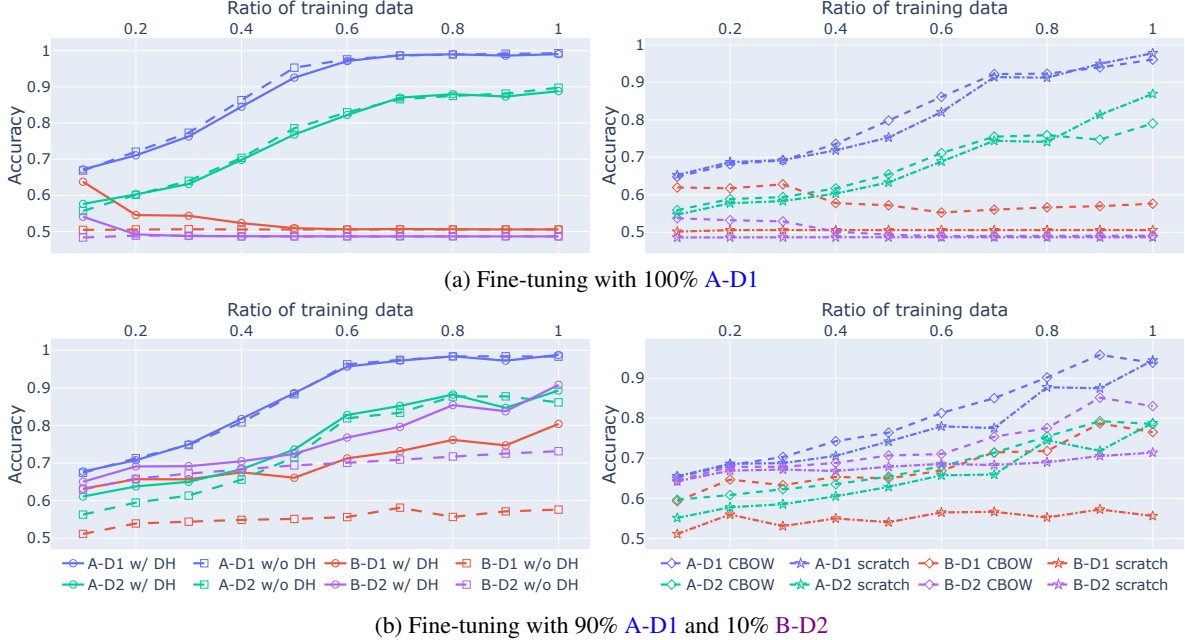

(a) Fine-tuning with 100% A-D1

- A-D1 w/ DH    A-D1 w/o DH    B-D1 w/ DH    B-D1 w/o DH
- A-D2 w/ DH    A-D2 w/o DH    B-D2 w/ DH    B-D2 w/o DH
- A-D1 CBOW    A-D1 scratch    B-D1 CBOW    B-D1 scratch
- A-D2 CBOW    A-D2 scratch    B-D2 CBOW    B-D2 scratch

(b) Fine-tuning with 90% A-D1 and 10% B-D2

Figure 3: Performance on the synthetic downstream task described in §4.2.

Our experiments also shed light on the effect of non-contextualized word embeddings and the statistics of parameter initialization. We find that the distributional property (Eq. 1) indeed contributes to non-contextualized embeddings' better sample efficiency and better generalization capability. As for the statistics of the parameter initialization, we find that models with shuffled weights marginally outperform models trained from scratch for all the settings. It suggests the statistics of parameters may not explain the efficacy of pretraining.

Additionally, we also observe phenomena that the distributional property (Eq. 1) cannot explain. We find that the distributional property is unrelated to the semantic distribution shifts. We also observe that using CBOW embeddings is not as stable as using an MLM pretrained model. Therefore, the distributional property alone does not fully explain the efficacy of model pretraining.

As for the multi-token setting, sometimes the distribution property does not improve sample efficiency but still helps generalization. We include the results in A.2 for brevity.

## 6 Analyzing Models for Real-world Datasets

In § 5, we show that the distributional property in Eq. 1 can improve sample efficiency, at least when the feature is at the single-token level. We posit that this conclusion applies to the real-world scenario

where the data also has this property. However, the negative results regarding the generalization capability may be due to the oversimplification of our toy setting. Therefore, in this section, we study generalization in real-world scenarios.

Unfortunately, it is difficult to conduct a fully controlled experiment as we do with the toy settings. It is impossible to find out all the synsets in a natural language. It is also difficult to divide features into isomorphic subsets as the feature sets $\Phi_a$ and $\Phi_b$ we have in the toy settings. Therefore, we will measure the relationship between the distributional property in Eq. 1 and generalization in an indirect manner.

### 6.1 Experiment Design

We want to measure to what degree we can attribute models' generalization capability to the distributional property (Eq. 1) of the pretraining data. Thus, we design an experiment using the following premise: if a fine-tuned model $f$ generalizes well because the pretrained model can learn semantic equivalence between features by modeling the distribution property in Eq. 1, then whether $f$ can generalize should depend on whether $f_0$ models Eq. 1 successfully.

Based on this premise, we design our experiment in the following way:

- Given a validation or testing sample $x$ in a downstream task, we pick a feature $a$ from $x$

and generate a noisy paraphrase $b$ (§6.2).

- We measure how well the fine-tuned model $f$ generalizes by how much its prediction changes after we replace $a$ in $\boldsymbol{x}$ with $b$:

$$D_f(\boldsymbol{x}, a, b) := \text{TV}[f(\boldsymbol{x}) \| f(\text{Replace}(\boldsymbol{x}, a, b)],$$

where TV is the total variation distance.

- We also measure whether the pretraining process encodes the semantic equivalence between $a$ and $b$ in the model, which we will elaborate on later in §6.3.

- Finally, we measure the Pearson correlation coefficient between $D_{f_0}(a, b)$ and $D_f(x, a, b)$. We should observe that $D_f$ is high only when $D_{f_0}$ is high if the fine-tuned model generalizes totally based on the semantic equivalence encoded by the distributional property in Eq. 1.

We analyze a `bert-base-uncased` model for a single-sentence and a multi-sentence task, the SST-2 task, and the MNLI task. The MNLI dataset comes with its constituency parsing along with the POS tags. As for the SST-2 dataset, it does not include the POS tags, so we parse it with the Stanford parser (Manning et al., 2014). In addition to analyzing the final fine-tuned model, we report the results for intermediate checkpoints. We also analyze models trained with different amounts of data to inspect the effect of training data size.

## 6.2 Noisy Feature Paraphrases

We use validation and test examples in this analysis and extract features at different levels: word level, phrase level, sentence level, and example level. For word-level features, we first measure the word saliency (Li et al., 2016; Ren et al., 2019)[3] and extract the word in the example with the highest saliency. For the phrase-level features, we consider both short phrases and long phrases in an example. For short phrases, we measure the saliency of phrases no longer than 4 words and choose the one with the greatest saliency. For long phrases, we choose the longest phrase in the example. A sentence-level feature is a whole sentence in a testing sample, while an example-level feature may involve more than one sentence depending

---

[3]i.e., we measure the impact of a feature by measuring the change of the fine-tuned model's prediction after we mask the feature.

on the task. For example, in the NLI task, each example involves two sentences.

We use WordNet (Miller, 1998) and back-translation (Sennrich et al., 2016; Yu et al., 2018) to generate semantically similar features. For a word-level feature $a$, we randomly select a word $b$ from one of its synonyms in WordNet that has the same POS tag. For a feature at a higher level, we use back-translation to paraphrase it because we find that back-translation generates more diverse paraphrases than publicly available paraphrase models do. We use Facebook FAIR's WMT19 translation models (Ng et al., 2019) and use German as the intermediate language. When generating a paraphrase $b$ for a feature $a$ in $x$, we use the context before $a$ in $x$ as the prefix of the generation process, so $b$ has the same syntax role as $a$. We include some examples in Table 1.

Note that $a$ and $b$ do not need to be perfectly similar. Although an imperfect $(a, b)$ pair may cause the fine-tuned model to change its prediction (having high $D_f(x, a, b)$), a pretrained model should also identify $a$ and $b$ as dissimilar (having high $D_{f_0}(a, b)$). Therefore, noises in our paraphrasing generation process do not affect our analyses.

## 6.3 Measuring Semantic Distance from a Pretrained Model

If pretrained models learn the semantic relationship between features by modeling the distributional property in Eq. 1 during MLM pretraining, then the distribution it predicts will reflect its knowledge about the semantic relationship. Specifically, if MLM pretraining encodes the semantic equivalence between two features $a, b$ in a pretrained model $f_0$, then the model should predict similar context distributions for $a$ and $b$. Therefore, we can check whether the pretraining process encodes the semantic equivalence between feature $a$ and $b$ by

$$D_{f_0}(a, b) := \text{TV}[f_0(\text{context}|a) \| f_0(\text{context}|b)],$$

where TV is the total variation distance.

We use templates to construct queries for the context distribution conditioned on a feature. A desirable template should convert a feature into a grammatical sentence containing a mask token such that the distribution at the mask token can reflect the semantics of the feature. Therefore, we design different templates to take the features' syntax role into consideration:

**Templates for word-level and phrase-level features** We expect that the verb after a noun can reflect its meaning because different things act in different ways. For example, both cats and dogs walk, but only dogs bark. Therefore, if the feature is a noun or noun phrase, then we use "`<feature> [mask]`" as its template. Based on a similar intuition, if the feature is an adjective or adjective phrase, then we use template "`[mask] is <feature>`". If the feature is a verb or verb phrase, we use the template "`[mask] <feature>`". Sentences with this syntax structure are pervasive in the pretraining data, so these templates construct natural sentences as the queries for the semantic relationship in the pretrained model.

**Templates for sentence-level features** We can see a sentence as a very high-level feature. For this kind of feature, we use two templates. The first one is more natural: "`<feature> with [mask].`", where we attach a prepositional phrase after the feature using "with". Because the word after "with" largely depends on the semantics of the context before it, the distribution at the "`[mask]`" reflects the semantics of the feature. We also experiment with a template proposed by Jiang et al. (2022) because they show that "`"<feature>" means [MASK]`" can extract high-quality sentence representations.

**Templates for example-level features** At the highest level, we can treat an example of a task as a feature. That is, for tasks that involve multiple sentences, such as a natural language inference task, we treat a pair of sentences as a feature. For this high-level feature, we use task-specific templates from Gao et al. (2021). These templates can elicit the model's prior knowledge about the task. We will inspect to what extend such prior knowledge is preserved after we fine-tuned the model.

### 6.4 Results and Discussion

Figure 4 shows that the correlation between $D_{f(x,a,b)}$ and $D_{f_0}$ is below 0.15 for both MNLI and SST-2. For MNLI, higher-level features have a higher correlation. As for SST-2, task-level features still have the highest correlation, but short-phrase-level features have the second-highest correlation. It may be because the SST-2 model utilizes lower-lever features more. We also observe that the correlation does not diminish to 0 regardless of the training steps or data sizes. It suggests that the real-world scenario may be more similar to the high-level toy setting where isomorphic feature sets

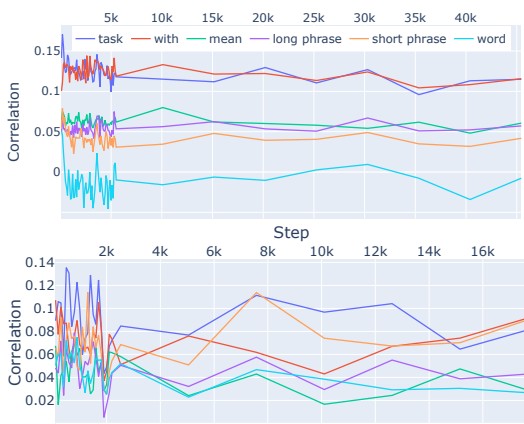

(a) MNLI (up) and SST-2 (down) at different steps. We measure the correlation per 100 steps at the beginning.

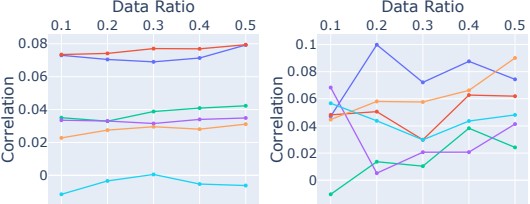

(b) MNLI (left) SST-2 (right) with different data sizes.

Figure 4: Correlation between $D_{f(x,a,b)}$ and $D_{f_0}$.

share the vocabulary (in Appendix A.2). However, in general, the correlation is low, indicating that the model's knowledge learned from the distributional property (Eq. 1) of the pretraning data can hardly explain models' generalization.

## 7 Related Work

This work aims to inspect how MLM pretraining infuses inductive bias for downstream tasks. It is different from previous studies that treat pretrained models as static models. For example, Wei et al. (2021) and Chiang (2021) study the efficacy of MLM theoretically by analyzing the distribution predicted by the model. Tenney et al. (2019b,a); Liu et al. (2019a); Hewitt and Manning (2019); Wu et al. (2020); Zhang et al. (2021) study the rich linguistic structure encoded in the representation. Some works focus on the fine-tuning process but the pretraining loss. For instance, Hao et al. (2019) shows MLMs have better loss landscape. Aghajanyan et al. (2021) shows that pretrained models have lower intrinsic dimensionality. Malladi et al. (2022) shows neural tangent kernels (Jacot et al., 2018) can explain MLM's efficacy to some extent. Some works study the connection between pretraining and downstream tasks. Li et al. (2021) show that non-language pretraining tasks help NLP

downstream tasks, while Lu et al. show that language models help non-language downstream tasks. Lovering et al. (2021) uses some linguistic tasks to show that the linguistic structures in MLMs can serve as an inductive bias. Zhang and Hashimoto (2021) investigate whether MLMs' efficacy can be explained by the task-relevant cloze-like masks alone. Our work follows this line and explains MLMs' general machine-learning characteristics.

Beyond the scope of NLP, the concept of connecting training examples is very common in machine learning. For example, co-training (Blum and Mitchell, 1998) uses multiple views of features to build connections between training samples. Recently, Saunshi et al. (2019); HaoChen et al. (2021) show that contrastive self-supervised learning reduces sample efficiency by pulling the representation of similar samples together. Similarly, Shen et al. (2022) shows that contrastive learning connects samples in different domains and thus helps generalization. However, those studies mainly focus on contrastive learning for continuous data instead of discrete languages.

## 8 Discussion

We showed in a synthetic experiment that pretraining with a masked language model objective allows the model to make use of semantic equivalence that simple distributions (e.g. Markov models) can encode to improve models' sample efficiency. However, we found that it can not explain generalization in the presence of distribution shifts.

Our work leads to two future research directions. First, we showed the limitations of our understanding of pretraining. What semantic relationships are important, how data distributions encode those relationships, and how modeling data distributions help downstream performance remain open questions. Second, our results in §6 show the unpredictability of models' generalization behavior. Investigating this behavior may help us develop more robust NLP models.

## Limitations

Our main goal is to provide a new perspective to analyze the efficacy of pretraining and to show that the distributional hypothesis is not a sufficient explanation. We only study semantic equivalence between features. Other relationships, such as hypernyms or causal relationships, are not in the scope of this paper. Additionally, we only consider

a relatively simple case, where tasks are pattern-matching tasks. We do not consider, for example, tasks that require reasoning. In §4, we study the effect of the distributional property using Markov distributions. It only suggests that model can achieve better sample efficiency by utilizing the semantic relationship encoded in this simple first-order distribution. How the models are able to utilize semantics encoded in more complicated distributions, such as the distribution of natural languages, remains unknown. In §6, we only consider the distribution of a single context token; however, the semantics of a feature may be encoded in the distribution in a subtler way. Finally, we show the inefficiency of the distributional hypothesis as an explanation by providing counterexamples with small language models. We leave the study of large language models for future work.

## Acknowledgements

We appreciate Joshua Robinson for providing valuable feedback on this paper. We also thank the members of my lab, tamagotchi lab at USC, for helpful discussions. Finally, we thank the USC NLP cluster maintenance team.

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

# A  Toy Experiments in the Multi-token Setting

## A.1  Independent Experimental Variables

**Feature Levels.** We can also control whether the semantic relationship in this pseudo language is at a single-token level or at a multi-token level, namely whether each feature corresponds to one or more tokens. This multi-token setting is interesting because the distribution of the token sequences will have higher-order dependence than a Markov chain. Additionally, multi-token features also make this pseudo-language more similar to natural languages.

We decide the mapping between features and token sequences before we start generating passages. We use two vocabulary sets $V_\alpha$ and $V_\beta$ for the feature sets $\Phi_a$ and $\Phi_b$ respectively. In the single-token setting, each feature in $\Phi_a$ and $\Phi_b$ is bijectively mapped to a token in $V_\alpha$ and $V_\beta$ respectively. In the multi-token setting, each feature in $\Phi_a$ and $\Phi_b$ corresponds to 1 to 3 randomly selected tokens from $V_\alpha$ and $V_\beta$ respectively.

**Vocabulary sharing between $\Phi_a$ and $\Phi_b$.** When the features are multi-token, $V_\alpha$ and $V_\beta$ can be the same while keeping the mapping between features and token sequences bijective. Therefore, we can choose whether to share the vocabulary between $\Phi_a$ and $\Phi_b$. The no-sharing setting is analogous to the multilingual setting in the real world, while the vocabulary-sharing setting is more similar to the monolingual setting.

## A.2  Results on the Multi Token Setting

### A.2.1  Sample Efficiency.

For the multi-token-level setting, Eq. 1 helps the sample efficiency only when the feature sets $\Phi_a$ and $\Phi_b$ have separated vocabulary. In Figure 5b, we can see the w/ DH model consistently outperforms the w/o DH model when there is no vocabulary sharing. However, the improvement is not as apparent as in the single-token-level setting. When the feature sets $\Phi_a$ and $\Phi_b$ have shared vocabulary, as shown in Figure 5a, the w/o DH model performs better than the w/ DH model. The results

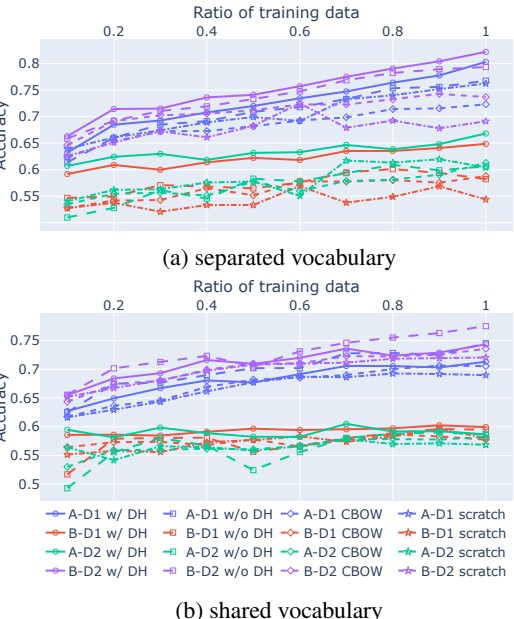

(a) separated vocabulary

(b) shared vocabulary

Figure 5: The downstream performance of fine-tuning with 50% A-D1 and 50% B-D2 in the multi-token feature setting.

indicate that MLM pretraining is not always able to infuse the inductive bias for the muti-token semantic relationship. As for the model initialized with CBOW embeddings, it barely outperforms the model trained from scratch. It shows that non-contextualized embeddings are not able to capture the semantic relationship at a higher level.

### A.2.2  Generalization Capability

When $\Phi_a$ and $\Phi_b$ have separated vocabularies, we have similar observations as in the single-token-level setting. When all the fine-tuned data is in A-D1, the w/ DH model generalizes to B-D1 and B-D2, and the generalization also diminishes when there is more data. As for the performance on A-D2, the w/ DH and w/o DH models outperform training from scratch too, though the advantage is not as apparent as in the single-token setting.

When 10% of the fine-tuning data is in B-D2, its generalization to B-D1 persists. However, w/ DH does not improve the generalization to B-D2 as much as in the single-token-level setting. Interestingly, the w/ DH model outperforms the w/o DH model for A-D2. It implies that the distributional property in Eq. 1 helps the generalization to semantic shift, which is different from the observations in the other settings. It is possibly because the w/ DH model has better sample efficiency. As for CBOW, it does not help generalization either, indicating that CBOW is not able to capture the multi-token

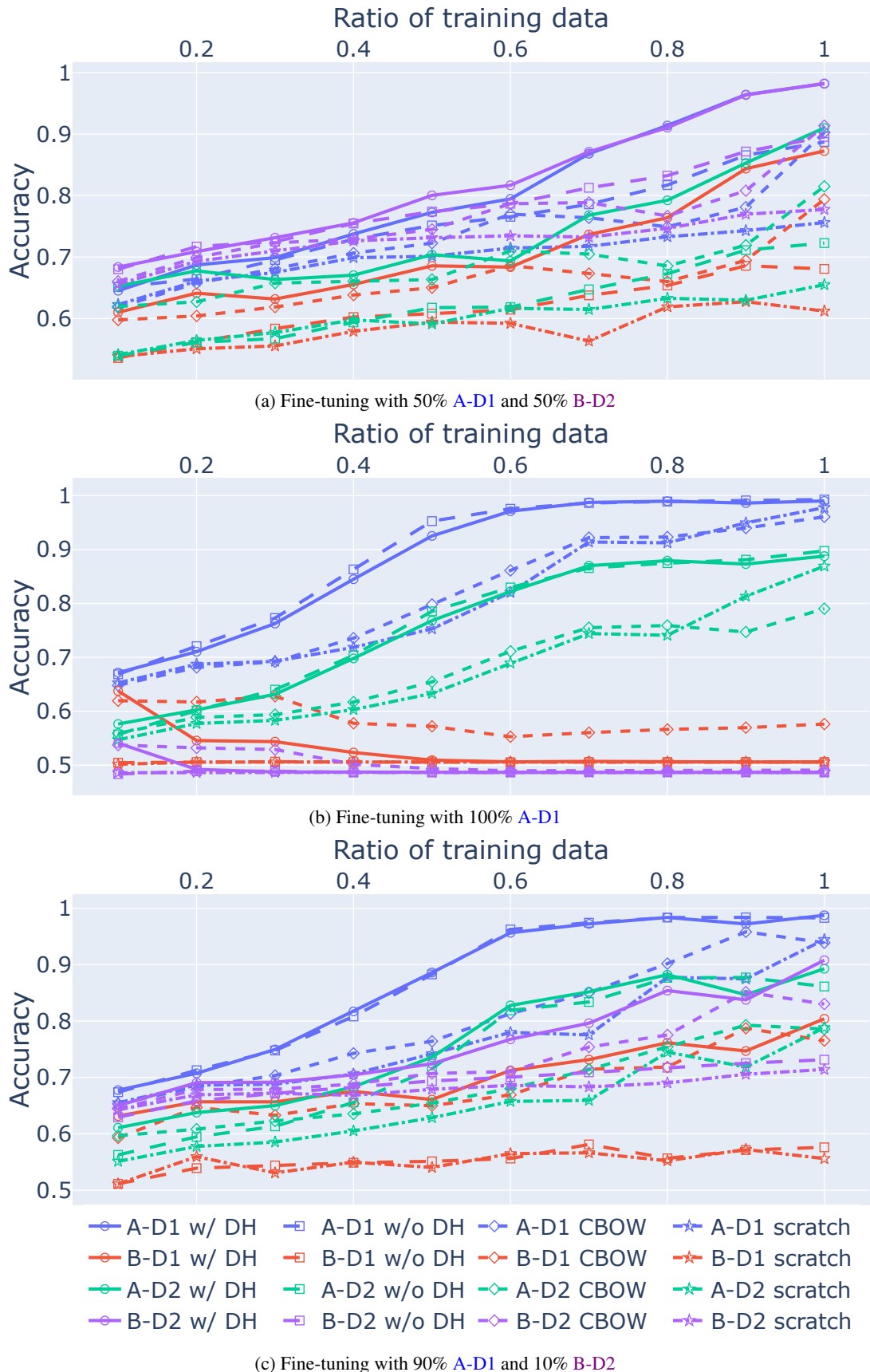

(a) Fine-tuning with 50% A-D1 and 50% B-D2

(b) Fine-tuning with 100% A-D1

(c) Fine-tuning with 90% A-D1 and 10% B-D2

Figure 6: Performance on the synthetic downstream task described in §4.2 (enlarged plots).

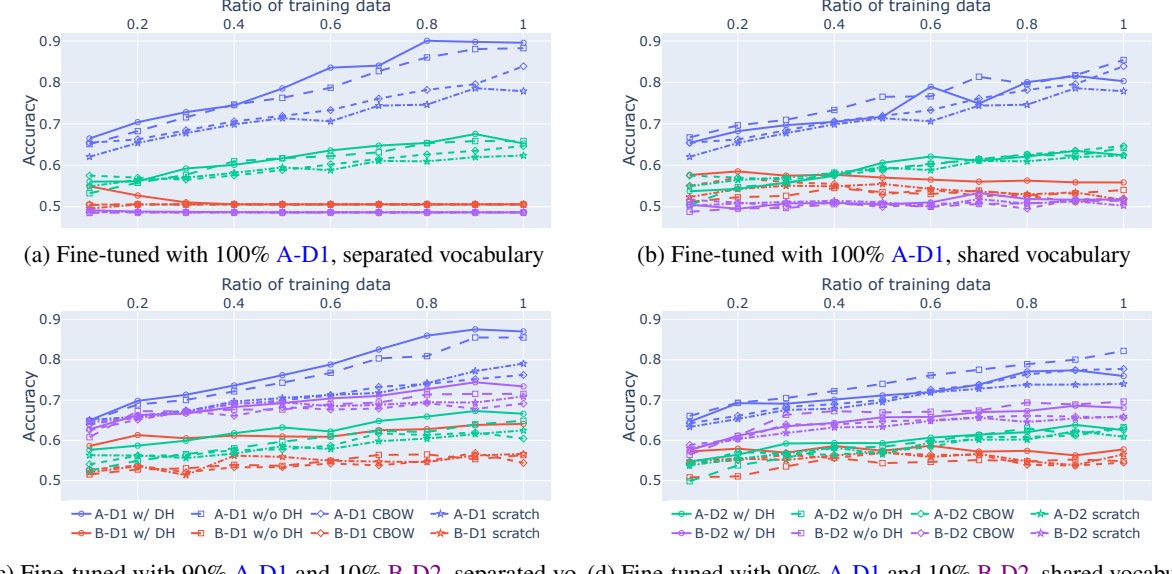

(a) Fine-tuned with 100% A-D1, separated vocabulary

(b) Fine-tuned with 100% A-D1, shared vocabulary

(c) Fine-tuned with 90% A-D1 and 10% B-D2, separated vocabulary

(d) Fine-tuned with 90% A-D1 and 10% B-D2, shared vocabulary

Figure 7: Performance of the downstream task in the multi-token feature setting.

semantic relationship.

When $\Phi_a$ and $\Phi_b$ share the vocabulary, we observe mixed results. We find that when all the fine-tuning data is in A-D1, the w/ DH model outperforms the w/o DH model on B-D1 persistently. It implies that, in this setting, the distributional propery in Eq. 1 helps the model generalize to vocabulary shift better than in the single-token setting. However, when 10% of the fine-tuning data is from B-D2, the advantage over the w/o DH model is less apparent. Additionally, regardless of whether we use data in B-D2 to fine-tune the model, Eq. 1 does not help the generalization to B-D2. MLM pretrained does not outperform the model trained from scratch on A-D2 either. It implies that MLM pretraining does not help the generalization to semantic shift in this setting.

## B Experiment Details For §5

We use a 6-layer transformer (Vaswani et al., 2017) with 12 heads. $P_\Sigma^{(1)}$ and $P_\Sigma^{(2)}$ are 2 fully-connected randomly generated Markov chains. Each node in the Markov chains has two outward edges pointing to two random nodes (as in Figure 1). Each node in a chain has uniform probability to be the starting node. For the pretrainng corpora, we sample 100k sequences, each of which contains 256 synsets. The dataset for the downstream task contains another 100k sequences, each of which contains 100 synsets. When fine-tuning the model, we early-stop when the validation performance does

not improve for 5 epochs consecutively. As for the labeling function, we use 5 patterns in total, and $|S_1| = |S_2| = |S_3| = 12$.

## C Experiment Details for §6

We use the Trainer API in the Python package transformers v4.21.2 with the default hyper-parameters. When doing back-translation, we use beam size equal to 5. Among the generated candidates whose score is not less than the score of the best candidates by more than 1, we choose the one with the highest edit-distance to the original input.

| Feature | Examples |
|---|---|
| Word | |

- the movie understands like few others how the depth and breadth of emotional **closeness (intimacy)** give the physical act all of its meaning and most of its pleasure.

- in an effort , i suspect , **not to spite (offend)** by appearing either too serious or too lighthearted , it spites by just being wishy-washy.

- if you are an actor who can relate to the search for inner peace by dramatically depicting the lives of others onstage , then esther 's story is a compelling **pursuit (quest)** for truth.

- dazzles with its fully-written characters , its determined stylishness ( which always relates to characters and story ) and johnny dankworth 's best soundtrack in **days (years)**.

- the title not only **draws (describes)** its main characters , but the lazy people behind the camera as well .

**Short phrase**

- if you are an actor who can relate to the search for inner peace by dramatically depicting **the lives of other people (the lives of others)** onstage , then esther 's story is a compelling quest for truth.

- the film is powerful , **approachable (accessible)** and funny.

- the best film about baseball to hit theaters since **"Field of Dreams" (field of dreams)**.

- a literate presentation that wonderfully weaves **a homicidal event (a murderous event)** in 1873 with murderous rage in 2002.

**Long phrase**

- although laced with humor and a few fanciful touches , the film is **a refreshingly earnest look at young women (is a refreshingly serious look at young women)**.

- **The three stakeholders of vera - mollà, gil and bardem - (vera 's three actors – mollà , gil and bardem –)** excel in insightful , empathetic performances.

- **coughs and stutters at his own post-modern vanity. (coughs and sputters on its own postmodern conceit .)**

- the old-world - meets-new mesh **is embodied in the soundtrack of the film, a joyous effusion of disco-Bollywood that lifted my spirit out of the theatre at the end of the monsoon wedding (is incarnated in the movie 's soundtrack , a joyful effusion of disco bollywood that , by the end of monsoon wedding , sent my spirit soaring out of the theater)**.

Table 1: Examples of the noisy paraphrase pairs used in Section 6. The text bold face parts is the noisy paraphrase (*b*) based on the original feature in the parentheses (*a*).

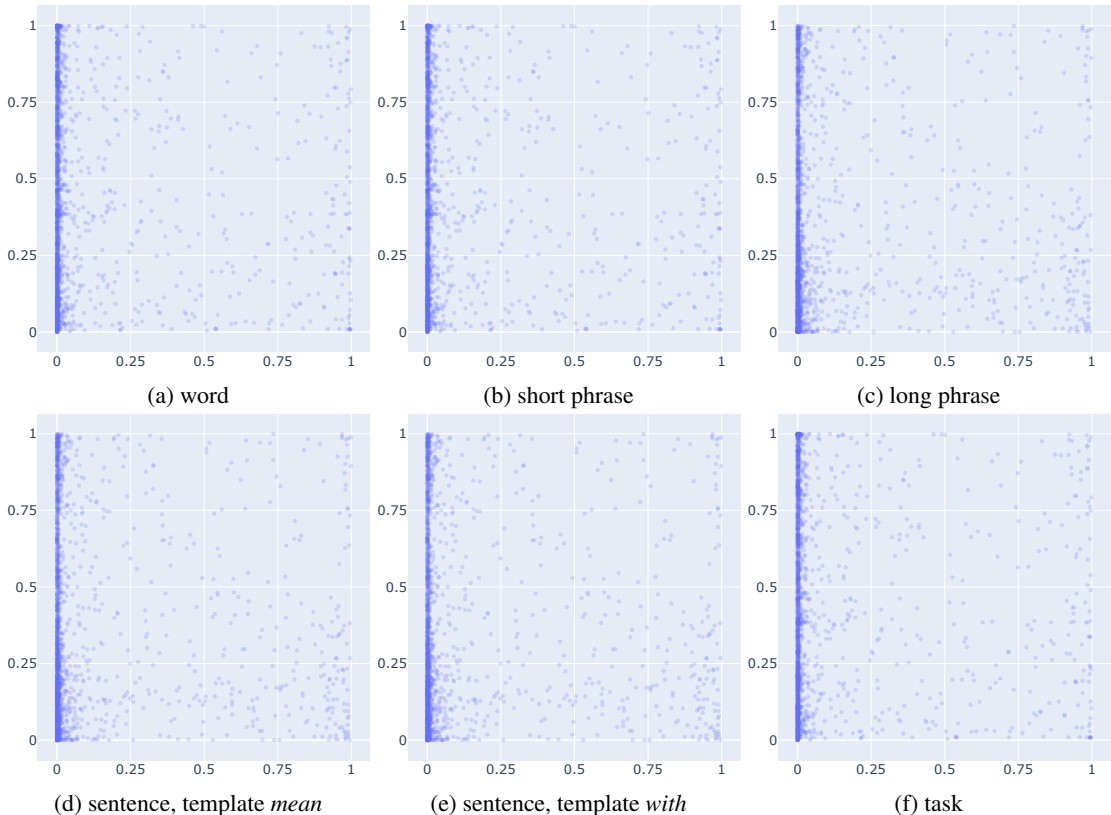

(a) word        (b) short phrase        (c) long phrase

(d) sentence, template *mean*     (e) sentence, template *with*     (f) task

Figure 8: Scatter plots of $\left(D_{f_0}, D_{f(x,a,b)}\right)$ for the MNLI experiment in §6, where the $x$-axis is $D_{f_0}$ and the $y$-axis is $D_f$.

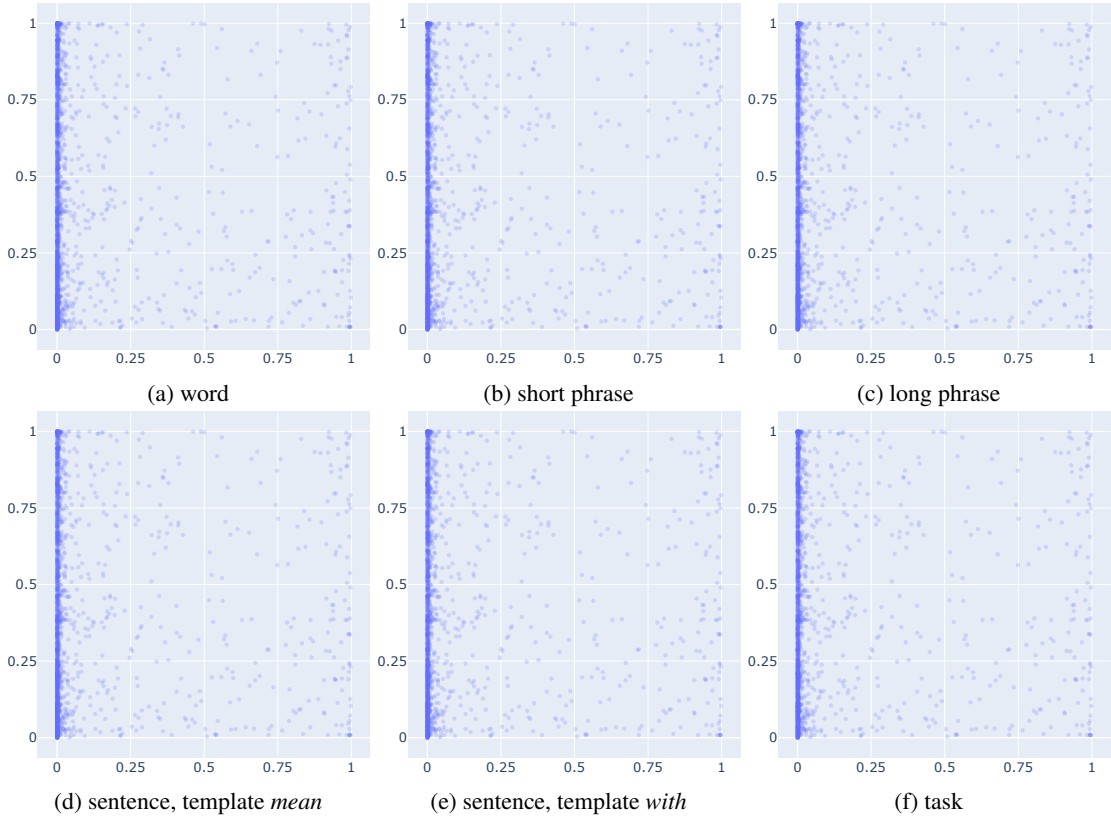

(a) word        (b) short phrase        (c) long phrase

(d) sentence, template *mean*     (e) sentence, template *with*     (f) task

Figure 9: Scatter plots of $\left(D_{f_0}, D_{f(x,a,b)}\right)$ for the SST-2 experiment in §6, where the $x$-axis is $D_{f_0}$ and the $y$-axis is $D_f$.