# OpenReview forum: "The Distributional Hypothesis Does Not Fully Explain the Benefits of Masked Language Model Pretraining"
_EMNLP/2023/Conference — EMNLP 2023 Main_

### Official Review · Reviewer_GJHm · 2023-07-26

**Soundness:** 2

**Excitement:**

3: Ambivalent: It has merits (e.g., it reports state-of-the-art results, the idea is nice), but there are key weaknesses (e.g., it describes incremental work), and it can significantly benefit from another round of revision. However, I won't object to accepting it if my co-reviewers champion it.

**Paper Topic And Main Contributions:**

This paper provides investigations on the distributional property of pre-trained language models. Evidence on synthetic data and real tasks is provided.

**Reasons To Accept:**

This work may be referable for readers in related line of research.

**Reasons To Reject:**

- The details of the synthetic data and the real tasks a not clear, which makes the analyses and figures difficult to grasp the meaning.
- The overall conclusion of the work is somehow negative. So that the work still has spaces to be improved.

**Reproducibility:**

3: Could reproduce the results with some difficulty. The settings of parameters are underspecified or subjectively determined; the training/evaluation data are not widely available.

**Reviewer Confidence:**

2: Willing to defend my evaluation, but it is fairly likely that I missed some details, didn't understand some central points, or can't be sure about the novelty of the work.

---

> ### Author Rebuttal · Authors · 2023-08-28
>
> We thank you for your comments.We will try our best to improve clarity. Could you please let us know which specific parts are unclear and how we can possibly improve our work? We will appreciate it if you can help us make our work better.

---

### Official Review · Reviewer_ggfY · 2023-08-01

**Typos Grammar Style And Presentation Improvements:** 1. Please mention which correlation m…
**Soundness:** 4

**Excitement:**

4: Strong: This paper deepens the understanding of some phenomenon or lowers the barriers to an existing research direction.

**Paper Topic And Main Contributions:**

This paper aims to analyze how the masked language model (MLM) pertaining helps downstream fine-tuning from the distributional hypothesis perspective. In particular, they analyze whether learning semantic equivalence leads to better sample efficiency and generalization capability.

To this end, they first design a well-controlled synthetic experiment. They show that the learned semantic equivalence during pretraining indeed improves sample efficiency during fine-tuning, but it does not fully explain the model's better generalization capability. Especially, it does not help when there is a semantic distribution shift.

Second, they extend the experiment to a real-data setting. They get the change of the fine-tuned and pre-trained models, respectively, when replacing a word/phrase/sentence/example with its paraphrase in the input.  Then they measure the correlation between these two changes. If they are highly correlated, then it means downstream finetuning directly makes use of the distributional knowledge learned in pretraining. However, they only observe low correlations.

Overall, they show that the distributional hypothesis, in particular, semantic equivalence, can partially explain the benefit of MLM pretraining but not fully.

The contributions of this paper are made toward both computationally-aided linguistic analysis and NLP engineering experiment directions.

**Questions For The Authors:**

Question A: This question is relevant to my 2nd reason to reject. As you mentioned in the limitations, the single-token context you considered in section 6.3 is quite limited. I wonder why not directly compare the contextualized embeddings of words/phrases/sentences from the pre-trained LM?

Question B: In A.2, do you think the conclusion will change if you do span-based MLM as opposed to the standard token-based MLM?

**Reasons To Accept:**

1. The synthetic experiment is very well-designed and performed. It clearly shows how MLM pretraining affects the sample efficiency and generalization of fine-tuning and will provide inspiration for future work in this direction. I especially like the way they caption vocabulary shift and semantic shift which is analogous to cross-lingual transfer and spurious correlations in real settings.

2. Although the results of the real-data experiment are less clear, the proposed method in this investigation is reasonable and can also inspire future work.

3. Analyzing how pretraining helps downstream fine-tuning is an interesting and important topic to study.

4. The paper is well-written and easy to follow.

**Reasons To Reject:**

1. Figure 2 is hard to read. Lines are cluttered, and I have to refer to the caption constantly to find a line that I want to see.

2. Compared to the synthetic experiment, the real-data experiment result is less clear and less informative. It is understandable that real-data experiment lacks control and thus are difficult to obtain clear conclusion from. Nonetheless, I think the experiments could have been done more thoroughly, e.g., try different ways of measuring the semantic distance from the pre-trained model.

**Reproducibility:**

4: Could mostly reproduce the results, but there may be some variation because of sample variance or minor variations in their interpretation of the protocol or method.

**Reviewer Confidence:**

4: Quite sure. I tried to check the important points carefully. It's unlikely, though conceivable, that I missed something that should affect my ratings.

---

> ### Author Rebuttal · Authors · 2023-08-28
>
> We would like to thank reviewer ggfY for their careful inspection of our work and the thoughtful comments. Below is our response:
>
> **Figure 2 is hard to read**
>
> We apologize that we sacrifice the accessibility of figures a little bit to get more space. We will try to make the figures easier to read with one additional page if our paper is accepted, e.g. enlarge the figures, include the caption in every subfigure. If feasible, we will also include figures in the appendix and each of them will only include the curves required for a specific comparison purpose.
>
> **Compare the contextualized embeddings**
>
> Thank you for this insightful question. We agree that comparing the contextualized embeddings is an intuitive way to infer the semantic equivalence learned by the model. However, compared with using the probability predicted by the model, the relationship between the contextualized embeddings and the data distribution is less clear. Because in this work we focus on understanding how modeling the data distribution is helpful, we prioritize using the probability predicted by the model in our experiment.
>
> **Using a span-based MLM in A.2**
>
> It is a very interesting question. We think it is possible that the conclusion may change if we use a span-based MLM. If time permits, we will run a few experiments and include the result in our later version.
>
> **Typos Grammar Style And Presentation Improvements**
>
> - 1: We use Pearson correlation. We will state it in our later revision.
> - 2-4: We will fix it!
> - 5: You are correct. We will proofread this section again.
>
> Thanks a lot for pointing out these issues!

---

### Official Review · Reviewer_wjUa · 2023-08-07

**Soundness:** 3

**Excitement:**

3: Ambivalent: It has merits (e.g., it reports state-of-the-art results, the idea is nice), but there are key weaknesses (e.g., it describes incremental work), and it can significantly benefit from another round of revision. However, I won't object to accepting it if my co-reviewers champion it.

**Paper Topic And Main Contributions:**

This paper analyzes the masked language modeling (MLM) pretraining objective function from the perspective of the Distributional Hypothesis. It investigates whether the improved sample efficiency and generalization capability of pretrained MLM models can be attributed to the semantic similarity encoded in the distributional property of the pretraining data.

The main contributions of the paper are as follows:

The paper conducts an analysis using a synthetic dataset to explore the influence of the distributional property on the sample efficiency of pretrained MLM models. The results suggest that the distributional property does contribute to better sample efficiency.

The paper also extends the analysis to two real-world datasets to assess the impact of distributional property on the generalization ability of pretrained natural language models. The findings indicate that the distributional property alone does not fully explain the models' generalization capability.

**Reasons To Accept:**

The strengths of this paper are as follows:

Novel Perspective: The paper adopts a novel perspective by analyzing the masked language modeling (MLM) pretraining objective function through the lens of the Distributional Hypothesis.

Synthetic Dataset Validation: By conducting experiments on a synthetic dataset, the paper offers a controlled environment to study the influence of distributional properties. This adds validity to the findings and helps establish causal relationships between distributional properties and sample efficiency.






**Reasons To Reject:**

The weaknesses of this paper are as follows:

Limited Scope: The paper explicitly states that it only considers a relatively simple case of pattern matching tasks, and semantic equivalence between features is the primary focus. This limited scope may restrict the generalizability of the findings to more complex NLP tasks or tasks involving different types of relationships between features.

Omitted Relationships: The paper explicitly states that it does not consider other relationships, such as hypernyms or causal relationships, between features. However, these relationships could be relevant to understanding the efficiency of pretraining and generalization in certain NLP tasks.



**Reproducibility:**

3: Could reproduce the results with some difficulty. The settings of parameters are underspecified or subjectively determined; the training/evaluation data are not widely available.

**Reviewer Confidence:**

4: Quite sure. I tried to check the important points carefully. It's unlikely, though conceivable, that I missed something that should affect my ratings.

---

> ### Author Rebuttal · Authors · 2023-08-28
>
> We thank Reviewer wjUa for their thoughtful comments. We agree with the reviewer that our work does not consider all semantic relationships. However, we would like to point out that
>
> 1. Even though the Distributional Hypothesis has been widely accepted as an explanation for the efficacy of non-contextualized word embeddings (Levy and Goldberg, 2014) and as a plausible hypothesis for the efficacy of MLM pretraining (Sinha et al., 2021), the meaning of “the Distributional Hypothesis” in this context (i.e. what semantic relationships can be encoded in the data distribution in what way) and how it is helpful for pretraining (i.e. among the encoded semantic relationships what relationships are helpful and in what way) has never been clearly stated. One contribution of our work is that we make a first step to formalize the statement. It is the well-defined scope of our formulation that allows us to scientifically inspect the hypothesis. Our efforts may inspire future exploration of other semantic relationships.
> 2. We theoretically and empirically analyze the benefit of modeling semantic equivalence. In particular, in Section 5, we show that semantic equivalence alone is enough to partially explain MLM pretrained models’ sample efficiency.
>
> Therefore, we believe that our work is beneficial for future researchers to better understand why MLM pretraining is beneficial.

---

### Official Review · Reviewer_GgVT · 2023-08-11

**Soundness:** 4

**Excitement:**

4: Strong: This paper deepens the understanding of some phenomenon or lowers the barriers to an existing research direction.

**Paper Topic And Main Contributions:**

The authors in this paper investigate whether mask language model pretraining allows models to use semantic knowledge extracted from the distribution of words in the pretraining data for downstream tasks. They show that prior knowledge about semantic equivalence alone is enough to result in better sample efficiency. Additionally, demonstrate the existence of a form of generalization capacity that is independent of the distributional property of the pretraining data. Furthermore, the distributional property of the pretraining data does not explain pretrained models'  generalization capability in the real world.

**Questions For The Authors:**

1. When replacing the words, how do you handle the case for ambiguous words? For example, the word “bass” can take the “musical instruments” or “a type of fish” meanings.
2. For semantic shift, do you also consider the semantic change of a word over time?


**Reasons To Accept:**

The paper is well-written and easy to follow. It highlights the boundaries of our understanding regarding models' pretraining prompting further examination of the intricate interplay between semantic relationships, data distribution and model pretraining to better understand the generalization behaviour of pretrained models.

**Reasons To Reject:**

I do not have a specific reason to reject the paper.

**Reproducibility:**

4: Could mostly reproduce the results, but there may be some variation because of sample variance or minor variations in their interpretation of the protocol or method.

**Reviewer Confidence:**

3: Pretty sure, but there's a chance I missed something. Although I have a good feel for this area in general, I did not carefully check the paper's details, e.g., the math, experimental design, or novelty.

---

> ### Author Rebuttal · Authors · 2023-08-28
>
> We are grateful for your recognition of our contribution. Regarding your questions, please see below our answers:
>
> **Q1: Handle the case for ambiguous words**
>
> For phrase-level and sentence-level features, because we use back-translation to generate their noisy paraphrases, we expect that the translation model is able to infer the correct word sense according to the context. As for word-level features, we do not take the multiplicity of word senses into consideration. However, through manually inspecting the samples, we do not find many ambiguous words in our noisy paraphrase/synonym pairs. We will include more examples in our later revision.
>
> **Q2: Consider the semantic change of a word over time for the semantic shifts**
>
> In our toy experiment, the relationship between the synsets ($\Sigma$) and features ($\Phi_a$ and $\Phi_b$) is static. If we see the synset of a feature as its semantic, we do not consider the semantic change of the words. However, we acknowledge that the semantic change of a word may cause the change of their occurrence distribution and thus cause a semantic distribution shift. This could be an interesting direction for future research!
>
> Thank you again for your insightful questions!

---

### Meta-Review · Area_Chair_mFAX · 2023-09-19

**Recommendation:** 4

**Metareview:**

The main contribution of this paper is formal definition of the "distributional hypothesis" which allows the authors to set up a synthetic dataset on which to measure the impact of this phenomena on language models. They conclude that this definition of semantic equivalence explains partially the models' sample efficiency (but does not fully explain their generalization capability).
As one could expect, a set-up which aims to measure this on real data is less conclusive which might be explained due to a myriad of cofounding variables.

This type of contribution is compelling due to its originality, and this was mentioned by the reviewers. The main drawbacks is the limited conclusions that can be drawn from such a synthetic set-up, something that is expected from a first work in this direction.

---

### Decision · Program_Chairs · 2023-10-07

**Decision:**

Accept-Main

**Comment:**

The main contribution of this paper is formal definition of the "distributional hypothesis" which allows the authors to set up a synthetic dataset on which to measure the impact of this phenomena on language models. They conclude that this definition of semantic equivalence explains partially the models' sample efficiency (but does not fully explain their generalization capability).
As one could expect, a set-up which aims to measure this on real data is less conclusive which might be explained due to a myriad of cofounding variables.

This type of contribution is compelling due to its originality, and this was mentioned by the reviewers. The main drawbacks is the limited conclusions that can be drawn from such a synthetic set-up, something that is expected from a first work in this direction.